# *Leishmania* Seroprevalence in Dogs: Comparing Shelter and Domestic Communities

**DOI:** 10.3390/ani13142352

**Published:** 2023-07-19

**Authors:** Paulo Afonso, Ana Cláudia Coelho, Hélder Quintas, Luís Cardoso

**Affiliations:** 1CECAV—Animal and Veterinary Research Centre, University of Trás-os-Montes and Alto Douro (UTAD), 5000-801 Vila Real, Portugal; afonso.p@icloud.com (P.A.); lcardoso@utad.pt (L.C.); 2Associate Laboratory for Animal and Veterinary Science (AL4AnimalS), University of Trás-os-Montes and Alto Douro (UTAD), 5000-801 Vila Real, Portugal; 3Agrarian School, Polytechnic Institute of Bragança (IPB), 5300-253 Bragança, Portugal; helder5tas@ipb.pt; 4Mountain Research Center (CIMO), Polytechnic Institute of Bragança (IPB), Campus de Santa Apolónia, 5300-253 Bragança, Portugal; 5Department of Veterinary Sciences, University of Trás-os-Montes and Alto Douro (UTAD), 5000-801 Vila Real, Portugal

**Keywords:** dog, domestic, leishmaniosis, Portugal, shelter

## Abstract

**Simple Summary:**

Shelter dogs are considered more susceptible to infection with *Leishmania infantum* than domestic dogs due to the living conditions they are subjected to. These two populations of dogs were compared in an area where leishmaniosis is endemic, and shelter dogs were found to be less infected than domestic dogs. Statistically significant differences were also found between age groups and clinical statuses. Monitoring, preventing, and treating canine leishmaniosis is crucial in reducing this zoonosis among animals and humans, under the scope of One Health.

**Abstract:**

Canine leishmaniosis (CanL) is a chronic, systemic, and often severe disease. The main causative agent of CanL is a protozoan parasite, *Leishmania infantum*, with phlebotomine sand flies acting as vectors. In Europe and other continents, *L. infantum* is also responsible for leishmaniosis in other animals, such as cats, horses, and humans. In Portugal, animal and human leishmaniosis is endemic, and high prevalence levels of infections and disease have been reported in dogs. There is a prejudice against stray animals and also those housed in shelters, assuming they have higher levels of infection with vector-borne pathogens, including *L. infantum*, when compared to domestic animals. In northeastern Portugal, serum samples were obtained from March to May 2022 in three shelters (*n* = 179) and thirteen veterinary clinics (*n* = 164), resulting in 343 dogs being analyzed for antibodies to *Leishmania* spp. by the direct agglutination test (DAT). The overall seroprevalence was 9.9%, with 15.2% seroprevalence in domestic dogs and 5.0% in the shelter ones (*p* = 0.003). The fact that shelter dogs had a lower seroprevalence could be explained by more regular veterinary care provided in shelters regarding preventive measures, including insecticides with an antifeeding effect, in comparison with domestic dogs.

## 1. Introduction

Although the infection can be subclinical, the disease canine leishmaniosis (CanL), caused by the protozoan parasite *Leishmania infantum*, is a chronic, systemic, and severe illness, which can often be fatal if not treated [1]. Apart from infecting dogs, which are its primary reservoir, *L. infantum* is also responsible for leishmaniosis in other animals, such as cats [2], horses [3], and humans [4], with phlebotomine sand flies acting as vectors [1]. Zoonotic visceral leishmaniosis (VL) is endemic in many geographic regions, including Europe, with a particular occurrence in southern Europe and the Mediterranean basin, where it is a primary veterinary and public health concern [5]. A review by Franco et al. described an overall seroprevalence of 23.2% (median: 10%) in 504,369 dogs from Italy, France, Spain, and Portugal tested between 1971 and 2006 [6]. Human VL caused by *L. infantum* is a notifiable disease in Portugal, with around 80 officially notified cases from 1999 to 2021 [6,7]. There has been a growing interest in understanding the seroprevalence of leishmaniosis in dogs housed in shelters [8,9,10,11,12] in order to understand their role in the epidemiology of CanL and to develop more effective control strategies [12]. CanL has been assumed to be particularly problematic in stray and sheltered animals, which may have poor living conditions and health and nutritional needs, and overcrowding could increase the risk of transmission associated with potentially fewer preventive measures [8,13,14]. Consequently, there is a prejudice against these animals, assuming they would have a higher prevalence of infection and disease than their domestic counterparts [12,13,14,15]. The prevalence and distribution of CanL have been studied in Portugal by several authors [13,14,16,17,18,19,20,21]. Recent results suggest a duplication of seroprevalence in domestic dogs over a 10-year time frame [13]. However, studies in Portugal have yet to include shelter animals, a circumstance that represents a knowledge gap and could lead to a biased evaluation of the prevalence of infection. The present work aimed to assess the seroprevalence of antibodies to *Leishmania* spp. in two groups of dogs living in northern Portugal, where CanL is endemic. In addition, the association between seropositivity and *Leishmania* and the potentially associated risk factors were evaluated.

## 2. Materials and Methods

### 2.1. Geography

This study was conducted in the district of Bragança, located in northeastern Portugal, which is part of the historical province of Trás-os-Montes e Alto Douro. This district is made up of 12 municipalities: Alfândega da Fé, Bragança, Carrazeda de Ansiães, Freixo de Espada à Cinta, Macedo de Cavaleiros, Miranda do Douro, Mirandela, Mogadouro, Torre de Moncorvo, Vila Flor, Vimioso, and Vinhais. The geographical area spans 6608 km^2^ and has a resident human population of 122,804 inhabitants, according to the 2021 census [22].

The region has a wide temperature range, with extreme values reaching 40 degrees Celsius in summer and minus 10 degrees Celsius in winter [23]. The total average rainfall ranges from 15.4 mm in July to 121.6 mm in December, not exceeding an annual average of 772.7 mm [23].

### 2.2. Animals and Samples

This study was approved by the Ethics Committee of UTAD (process reference: Doc6-CE-UTAD-2022). In addition, legal detainers or owners signed informed consent for the inclusion of dogs in this study.

A total of 179 dogs were sampled from the three official shelters (CRO, Official Collection Centre) of the district of Bragança. This represents the number of dogs available for sampling at the shelters. In addition, an equivalent number of domestic dogs was sampled at 13 veterinary medical centers (CAMV, Veterinary Medical Care Centre) covering 10 of the 12 municipalities: Alfândega da Fé (*n* = 1), Bragança (*n* = 3), Carrazeda de Ansiães (*n* = 1), Macedo de Cavaleiros (*n* = 1), Miranda do Douro (*n* = 1), Mirandela (*n* = 2), Mogadouro (*n* = 1), Vila Flor (*n* = 1), Vimioso (*n* = 1), and Vinhais (*n* = 1). The distribution of dogs per municipality was carried out in adjustment with the resident human population (according to the 2021 census [22]). A number of 164 domestic dogs were sampled from the veterinary medical centers.

The surplus of sera from blood samples collected in routine procedures at shelters and veterinary medical centers was used. Blood (1 mL) was collected by cephalic venipuncture, and serum was separated by centrifugation with samples transported under refrigeration. Serum was subsequently stored at a temperature of −20 °C until use. Data available on sex, age, breed, hair, habitat, clinical status (clinical manifestations related to CanL), municipality of origin, and use of ectoparasiticides (with action against phlebotomine sand flies) were registered for each dog (Table 1). All dogs were clinically examined for signs or manifestations compatible with CanL, including alopecia, anemia, anorexia, apathy/depression, arthritis/polyarthritis, ascites, muscle atrophy, cachexia, lameness, conjunctivitis, mucopurulent nasal discharge, skin desquamation, diarrhea, pain, epistaxis, splenomegaly, ulcerative stomatitis, glomerulonephritis, hematochezia, hepatomegaly, liver disease, cutaneous hyperpigmentation, hyperkeratosis, hyperthermia, jaundice, chronic renal failure, lymphadenopathy, melena, meningitis, cutaneous nodules, onychogryphosis, osteitis, panophthalmitis, weight loss/slimming, pyoderma, pneumonia, polyuria/polydipsia, purpura, keratoconjunctivitis sicca, renomegaly, rhinitis, cough, skin ulceration, uveitis, and vomiting [1].

### 2.3. Detection of Antibodies to Leishmania spp.

The direct agglutination test (DAT) was used to titrate IgG antibodies specific to *Leishmania* spp. based on a standard freeze-dried antigen at a concentration of 5 × 10^7^ promastigotes per milliliter in the direct agglutination test (DAT) to measure the titration of IgG antibodies specific to *Leishmania* spp. (Amsterdam University Medical Centers, Academic Medical Centre at the University of Amsterdam, Department of Medical Microbiology, Section Experimental Parasitology, Amsterdam, The Netherlands) [24].

Canine sera were serially diluted two-fold from 1:100 to 1:102,400 in saline solution (0.9% NaCl) containing 0.1 M β-mercapto-ethanol in V-shaped microtiter plates (Greiner, Germany). Plates were incubated at 37 °C for 1 h. After incubation, 50 uL of reconstituted DAT antigen was added to each well containing 50 uL of diluted serum. The positive control for DAT was a serum sample from a dog with leishmaniosis and a DAT titer ≥ 102,400. The negative control serum used was from a dog living in a geographical region where CanL is not endemic. Results obtained with DAT are expressed as an antibody titer, i.e., the reciprocal of the highest dilution at which agglutination (large diffuse blue mats) is still clearly visible after 18 h incubation at room temperature. To enhance sensitivity and specificity, a cut-off titer of 400 was selected [24].

### 2.4. Data Analysis

To compare the proportions of positivity among different categories of independent variables, the chi-square test and Fisher’s exact test (FET) were used, with a significance level set at *p* ≤ 0.05. In addition, exact binomial 95% confidence intervals (CI) were established for the values of general or overall seroprevalence and also for the values of seroprevalence related to the “origin” variable (i.e., shelter or domestic dogs). Stemstat, IBM SPSS Statistics 26^®^, and WinEpi software tools were used for the analyses.

## 3. Results

A total of 343 dogs were sampled, including the 179 dogs from shelters and the 164 domestic dogs. The age range was between 1 month and 19 years, with an average age of 55.9 months. With an expected prevalence of approximately 16% [13] and a confidence level of 95%, the estimated absolute error was approximately 5.4% and 5.6% for shelter (*n* = 179) and domestic dogs (*n* = 164), respectively. The sample population was sex balanced (Table 1).

The overall seroprevalence was 9.9% (CI: 6.7–13.1%). The prevalence was significantly different (*p* = 0.003) between domestic dogs (15.2%; CI: 10.1–21.7%) and shelter ones (5.0%; CI: 2.3–9.3%) (Table 1). When comparing results among different age groups, the prevalence was significantly different between young and adult dogs (0.0% (CI: 0.0–5.7%) versus 10.6% (CI: 6.7–15.8%); *p* = 0.045) and between young and senior dogs (0.0% (CI: 0.0–5.7%) versus 15.2% (CI: 8.1–25.0%); *p* = 0.009) (Table 1). The seroprevalence of antibodies to *Leishmania* was significantly different between apparently healthy (8.6%; CI: 5.7–12.2%) and sick dogs (25.0%; CI: 10.7–44.9%) (*p* = 0.013) (Table 1). Furthermore, no statistically significant differences were found between the categories of the independent variables, namely, sex, breed, habitat, use of ectoparasiticides, hair, municipality, and vaccination. (Table 1).

Seropositivity results by titer are presented in Table 2. The majority of shelter dogs (8/9) presented titers equal to or below 3200, with only one dog presenting a titer above 3200, i.e., of 51,200. Domestic dogs had a wide range of titers, with most (16/25) revealing titers equal to or greater than 6400.

## 4. Discussion

This study is an inaugural epidemiological investigation conducted on *Leishmania* infections in shelters in northern Portugal and is also the most extensive study carried out on the subject in the entire country. A lower seroprevalence was found in shelter dogs compared to domestic dogs. Despite our hypothesis considering shelter dogs more prone to infection and disease, our results revealed discordant results. A significantly different seroprevalence was found in adults and in senior dogs and sick dogs.

The few studies that have compared shelter to domestic dogs mostly found a higher seroprevalence in shelter dogs [8,12,14]. In Portugal, few studies have compared these two populations of dogs regarding seroprevalence. Cortes et al. analyzed 374 dogs in the Lisbon urban area between December 2002 and December 2003 with IFAT and found a seroprevalence of 18.4% (51/277) in domestic dogs and 21.6% (21/97) in shelter dogs with no statistical difference (*p* = 0.48) [14]. In Argentina, the seroprevalence of antibodies to *Leishmania* was significantly higher in shelter dogs (38.6%) compared to domestic dogs (20.1%) [8]. In Brazil, a survey conducted in 17 shelters found a seroprevalence of 33.7% (211/627) in sheltered dogs, ranging from 25.0% to 41.2%, contrasting with 3.4 to 9.6% found in previous studies involving domestic dogs [12].

Few studies have revealed results similar to those of the present report [25,26]. Colella et al. found a seroprevalence of 31.6% in domestic dogs contrasting with 14.6% in shelter dogs [26]. On the other hand, Tamponi et al. found a higher seroprevalence in domestic dogs (27.2%) than in shelter dogs (10.6%) [25]. 

Considering the limited number of national studies conducted [13,17], we prioritized using DAT to make our results comparable to those studies. We sought international studies [8,12,25,26] to achieve a broader comparative perspective, particularly in the context of shelters. Although these studies employed different serodiagnostic methodologies, the differences observed likely stem from real variations attributable to endemicity and other risk factors rather than inherent disparities due to the different methodologies used.

While some authors have not thoroughly examined the causes of seroprevalence variation between populations of shelter and domestic dogs, there are several possible explanations that have been proposed for the differences. Possible reasons for the higher seroprevalence of leishmaniosis in shelter dogs compared to domestic dogs include the lack of preventive measures [8], more favorable conditions for ectoparasite growth, due to organic material and blood meals [12], and limited access to veterinary care [27]. 

In the opposite direction, the higher seroprevalence in domestic animals may be explained by the sedentary lifestyle of domestic dogs and the propensity to be bitten by vectors and consequently infected [25]. In contrast, shelter animals are more likely to receive prophylactic measures due to the commitment of animal shelters compared to what occurs in domestic animals, as no preventive treatment was reported in 15.2% of domestic dogs in Sardinia (Italy) [25]. Furthermore, the lack of veterinary care and failure to use ectoparasiticides have been associated with higher seroprevalence levels in domestic dogs [27].

Anecdotally, the lower and higher seroprevalence numbers of *Leishmania* infection in shelter animals have been attributed to the use [25] or absence of prophylactic measures [8]. However, it is essential to note that these levels may reflect the context of each shelter and the specific application of preventive measures rather than a broad generalization that is applicable to all shelter animals.

Other studies have focused only on studying seroprevalence in shelters. In Italy, a seroprevalence of 5.0% was detected in a canine shelter [11]. In another study conducted in kennels, *L. infantum* had a prevalence of 2.5% [28]. In another study in Italy, seroprevalence levels of 1.8% and 10.0% were found in two shelters [10]. The authors point to the differences in environment to justify the lower prevalence in the former shelter, which is in a windy area with an absence of ravines and dry-stone walls. Despite the presumption that a shelter is a more suitable biotope for the vector and, in consequence, a place more susceptible to infection, no information about preventive measures (including ectoparasiticides) was given.

In the present study, dogs older than 12 months showed a higher seroprevalence (Table 1), discordant with the bimodal age distribution of the disease that suggests a higher prevalence in younger and older dogs [29,30,31]. However, the present study was in accordance with others that claim that the risk of infection rises with age [27,28,32,33]; the main explanation for this may be the outdoor lifestyle that older dogs have, which increases possible contact with vectors [27]. 

The prevalence of *Leishmania* was significantly different between apparently healthy (8.6%) and sick dogs (25.0%) (*p* = 0.013) (Table 1), with apparently healthy dogs representing 79.4% (27/34) of seropositive dogs. Sauda et al. found contrary results, with a higher representation of clinically suspected dogs at 62.5% (10/16) [28]. Seemingly, healthy dogs represent animals at risk of contracting infection and suffering from disease and also a reservoir of *Leishmania*, with potential transmission to other animals and humans, making early detection crucial [34]. In the present study, of those dogs with clinical signs (7/34; 20.6%), the most frequently detected manifestations were apathy/depression (*n* = 2), arthritis/polyarthritis (*n* = 2), lameness (*n* = 2), and onychogryphosis (*n* = 2), a situation which is in line with Otranto et al. [15]. Other clinical manifestation less observed were localized alopecia, anemia, anorexia, diarrhea, hepatopathy, lymphadenopathy (localized and generalized), melena, mucopurulent nasal discharge, muscular atrophy, nasal hyperkeratosis, pain, rash, skin desquamation, skin ulceration, uveitis, and vomiting.

In the present study, most shelter dogs had lower titers, a circumstance which contrasts with the higher titers in domestic dogs. There is a turning point with a reversal of frequencies starting from 6400. Higher titers may reveal a stronger but non-protective immune response [29,35]. Some authors consider that antibody titers are related to the clinical stage once it depends on the host’s immunologic response and relies on clinical signs, clinical–pathological abnormalities, and serologic status that are linked to the levels of antibodies for *L. infantum* [29,35,36].

The wide variety of clinical manifestations can be explained by the broad spectrum of manifestations that the disease can present [34]. Also, subclinical infection with *L. infantum* is more common than clinical disease in areas of endemicity [1]. Although most dogs infected with *Leishmania* spp. appear healthy or show no evident clinical signs [13], some of them can still transmit the parasite to the phlebotomine sand flies [37]. This situation perpetuates the *Leishmania* life cycle and puts humans at risk of infection, making leishmaniosis a significant veterinary and public health concern [38] and making an early diagnosis of *L. infantum* infection mandatory for correct management [39].

No statistical difference was noted between sex, breed, habitat, use of ectoparasiticides, hair, municipality, and vaccination (Table 1), suggesting a uniform distribution among the surveyed populations.

Several factors may contribute to the presence and spread of leishmaniosis in the Bragança district. One of the most important factors is the climate, which is characterized by hot and dry summers and mild winters [23], providing suitable conditions for the survival and reproduction of sand flies [18,40]. Another factor is the presence of wildlife reservoirs, such as foxes and rodents [41]. They can serve as sources of infection for sandflies [42], and contact with stray dogs can occur prior to their entrance into shelters.

The current climate change with increasing temperature and humidity contributes to the northward expansion of the expansion of the vector niche [43,44,45]. In addition, the rural landscape of Bragança district, with its traditional agricultural practices and livestock farming, can also contribute to the spread of leishmaniosis. Domestic dogs living in rural areas may be more likely to come into contact with sand flies, as they might be more exposed to these insects’ natural habitats [46], such as forests, riverbanks, or farms. Moreover, dogs used for hunting or herding may have a higher risk of exposure and infection as they spend more time outdoors and are more likely to get bitten by sand flies [34].

Finally, the lack of effective preventive measures, such as regular veterinary care and vaccination programs, may also contribute to the prevalence of leishmaniosis in the Bragança district. As a result, it is essential for dog owners, veterinarians, and public health officials to be aware of the risks associated with this disease and to take appropriate measures to prevent its spread. These measures could include the use of insect repellents, the implementation of vaccination programs, and the promotion of good hygiene and sanitation practices.

Official shelters are entities that generally provide a high standard of veterinary care to the animals they foster, including preventive measures to control infectious diseases. These measures include vaccinating the animals and implementing rigorous hygiene protocols to reduce the spread of diseases. Therefore, adopting an animal from an official shelter can be a safe choice, as these entities usually follow high standards of quality in relation to the health and welfare of the animals they shelter.

By adopting an animal from a shelter, especially an official one, people can have greater confidence that they are receiving an animal that is in good condition and in good health.

## 5. Conclusions

The study is the first of its kind in the region and is representative of the canine population density of the municipality in the area. This study suggests that the seroprevalence of *Leishmania* infection among domestic dogs has doubled in the last 10 years in the Bragança district. Furthermore, this study contradicts assumptions about higher seroprevalence in shelter/stray dogs compared to domestic dogs. It suggests that access to veterinary care provided in shelters, including prophylaxis against leishmaniosis, may be a contributing factor. The present study found a significant difference in the seroprevalence between shelter and domestic dogs, which challenges previous assumptions about higher seroprevalence in shelter and stray dogs due to inadequate living conditions, poor health, and overcrowding. Overall, this study suggests the importance of regular screening, prevention, and even treatment of leishmaniosis in both sheltered and domestic dogs, as they play a crucial role in transmitting *L. infantum* to other animals and humans. In addition, by improving dogs’ management and health, we can promote public health and well-being in their communities.

## Figures and Tables

**Table 1 animals-13-02352-t001:** Seroprevalence of *L. infantum* infection in dogs from northeastern Portugal by sex, breed, age group, habitat, use of ectoparasiticides, hair, clinical status, origin, municipality, and vaccination.

Variable	Title	Dogs Tested (n)	Relative Distribution (%)	DAT-Positive (n)	Seropositive (%)	95% CI
Origin (*p* = 0.003)	Shelter	179	52.2	9	5.0	2.3–9.3
	Domestic	164	47.8	25	15.2	10.1–21.7
Sex (*p* = 0.721)	Female	182	53.1	17	9.3	5.5–14.5
	Male	161	46.9	17	10.6	6.3–16.4
Breed (*p* = 0.702)	Defined	111	32.4	12	10.8	5.7–18.1
	Mongrel	232	67.6	22	9.5	6.0–14.0
Age group (*p* = 0.016)	Young ^a,b^	63	18.4	0	0	0.0−5.7
	Adult ^a^	198	57.7	21	10.6	6.7−15.8
	Senior ^b^	79	23.0	12	15.2	8.1−25.0
Habitat (*p* = 0.092)	Access to outdoors	92	26.8	14	15.2	8.6−24.2
	Totally indoors	30	8.7	1	3.3	0.1−17.2
	Totally outdoors	221	64.4	19	8.6	5.3−13.1
Ectoparasiticides (*p* = 0.332)	No	58	16.9	8	13.8	6.2−25.4
	Yes	285	83.1	26	9.1	6.1−13.1
Hair (*p* = 0.959)	Long	44	12.8	4	9.1	2.5−21.7
	Medium	75	21.9	8	10.7	4.7−19.9
	Short	224	65.3	22	9.8	6.3−14.5
Clinical status (*p* = 0.013)	Apparently healthy	315	91.8	27	8.6	5.7−12.2
	Sick	28	8.2	7	25.0	10.7−44.9
Municipality (*p* = 0.854)	Medium	128	37.3	12	9.4	4.9−15.8
	Small	215	62.7	22	10.2	6.5−15.1
Vaccination (*p* = 1.0)	No	333	97.1	33	9.9	6.9−13.6
	Yes	10	2.9	1	10.0	0.3−44.5
Total	All	343	100	34	9.9	7.0−13.6

^a^ *p* = 0.045; ^b^ *p* = 0.009. Bonferroni’s correction has been incorporated by multiplying a previously significant pairwise *p*-value (0.015) by 3. Only statistically significant differences are shown for pairwise comparisons of age group categories (i.e., young and adult or young and senior, respectively).

**Table 2 animals-13-02352-t002:** Shelter (*n* = 9) and domestic (*n* = 25) seropositive dogs by titer of antibodies to *Leishmania* spp. as determined by the direct agglutination test.

Titer	Group	Seropositive Dogs (n)	Frequency among the Same Group (%)
400	Shelter	2	22.2
	Domestic	3	12.0
800	Shelter	4	44.4
	Domestic	2	8.0
1600	Shelter	1	11.1
	Domestic	2	8.0
3200	Shelter	1	11.1
	Domestic	2	8.0
6400	Shelter	0	0.0
	Domestic	3	12.0
12,800	Shelter	0	0.0
	Domestic	3	12.0
51,200	Shelter	1	11.1
	Domestic	5	20.0
≥102,400	Shelter	0	0.0
	Domestic	5	20.0

## Data Availability

The data presented in this study are available upon request from the corresponding author.

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
