# Peer review of "Leishmania Seroprevalence in Dogs: Comparing Shelter and Domestic Communities"

_animals, 2023, doi:10.3390/ani13142352_

Round 1
Reviewer 1 Report (Previous Reviewer 1)
The editors have completely addressed all my questions.
Author Response
REVIEWER 1
The editors have completely addressed all my questions.
Author’s response: we thank Reviewer 1 very much for her/his positive evaluation of our manuscript.
Reviewer 2 Report (New Reviewer)
Dear authors,
The manuscript is very well-written and the topic is important in light of public health and specially it is focus in the difference of leishmania infection between domestic and shelter dogs.Please find below several technical mistakes and minor suggestions.
L21 human leishmaniosis instead of leishmanioses
in Materials and Methods it is missing the procedure of blood extraction (vein) and the transport of the samples.
L198 Pronity it is probably a mistake
L264, L265 and L266 in addition,....spread of leishmaniosis. Is there any reference to support that fact?
Overall it looks good to me, Good luck
Author Response
REVIEWER 2
Dear authors,
The manuscript is very well-written and the topic is important in light of public health and specially it is focus in the difference of leishmania infection between domestic and shelter dogs.
Author’s response (AR): we thank Reviewer 2 for her/his positive evaluation of our manuscript.
Please find below several technical mistakes and minor suggestions.
L21 human leishmaniosis instead of leishmanioses
AR: now adapted to read as: “animal and human leishmaniosis is endemic”.
in Materials and Methods it is missing the procedure of blood extraction (vein) and the transport of the samples.
AR: a description was added: “Blood (1 ml) was collected by cephalic venepuncture, and serum was separated by centrifugation with samples transported under refrigeration. Serum was subsequently stored at a temperature of -20°C until use.”
L198 Pronity it is probably a mistake
AR: “pronity” has been replaced with “propensity”.
L264, L265 and L266 in addition,....spread of leishmaniosis. Is there any reference to support that fact?
AR: Cardoso et al. (2021) was cited. Numbered as reference 42.
Overall it looks good to me, Good luck
AR: thank you very much!
Reviewer 3 Report (New Reviewer)
Authors assess the seroprevalence of antibodies to Leishmania spp. in dogs living in northern Portugal.
The title indicates the aim of the manuscript and the abstract clearly indicates the work objective, methodology and result of the study.
The introduction is also well written.
The objectives of the study are of interest and are in line with the scope of the journal.
The manuscript is well organized.
The conclusions are consistent with the evidence and arguments presented.
The reference is appropriate.
Some suggestions are required, as follows:
- In materials and methods, describe the collection of blood samples;
- Moderate editing of English language required.
In my opinion, the manuscript could be accepted for publication after minor revision.
Moderate editing of English language required
Author Response
REVIEWER 3
Authors assess the seroprevalence of antibodies to Leishmania spp. in dogs living in northern Portugal.
The title indicates the aim of the manuscript and the abstract clearly indicates the work objective, methodology and result of the study.
The introduction is also well written.
The objectives of the study are of interest and are in line with the scope of the journal.
The manuscript is well organized.
The conclusions are consistent with the evidence and arguments presented.
The reference is appropriate.
Author’s response (AR): we thank Reviewer 3 for her/his positive evaluation of our manuscript.
Some suggestions are required, as follows:
- In materials and methods, describe the collection of blood samples;
AR: a description was added: “Blood (1 ml) was collected by cephalic venepuncture, and serum was separated by centrifugation with samples transported under refrigeration. Serum was subsequently stored at a temperature of -20°C until use”.
- Moderate editing of English language required.
AR: the manuscript has now been reviewed by Mr. Kai Diprose, a native speaker of English.
In my opinion, the manuscript could be accepted for publication after minor revision.
AR: thank you very much!
This manuscript is a resubmission of an earlier submission. The following is a list of the peer review reports and author responses from that submission.
Round 1
Reviewer 1 Report
Manuscript animals-2439812
This is a very nice seroprevalence study on canine leishmaniosis in northern Portugal. It includes two populations of dogs, shelter dogs and privately owned dogs. Perhaps a bit surprisingly, the shelter dogs had a significantly lower seroprevalence than the owned dogs. The authors, however, provide background information that may account for this finding, like the good veterinary care the dogs enjoy at these official shelters.
Overall, the manuscript is very well written and offers all the information needed for the reader to appreciate what was done. I suggest to include an additional table, namely with the titers obtained for the seropositive dogs, separately for dogs in shelters and privately owned dogs.
Then there are some typos:
line 15: should be "differences"
line 108: the "7" in the concentration should be superscript
Table 1, categories "Ectoparasiticides" and "Clinical status": remove the line "total", as it is given at the end of the table.
line 186: If I am not mistaken, the reference 25 was done in Sardinia, so perhaps replace "southern Italy" by "Sardinia (Italy)"
line 197: should be "The authors point to the ..."
line 200: should be "is a more suitable..."
line 200: I do not understand the meaning of this part of the sentence "...and in consequence places more susceptible to infection"
line 221: What do the authors mean with "purple"?
line 226: "an" should be "can"
line 237: please check the sentence "...which can serve as souces of infection ... into shelters." I think there are two thoughts mingled together, perhaps rephrase.
line 239/240: suggest to reword "contribute to the northward expansion"
lines 268-273: this is a very long sentence - consider splitting it in two.
Reviewer 2 Report
The authors investigated the Leishmania sp. infection by DAT in dogs from Bragança district, in Portugal. However, it is not possible to define an infection based on only one serological test. Infections are confirmed only with positive parasitological tests (including molecular) or, in the absence of these, at least two positive serological tests, which can be DAT, ELISA, IFAT, point-of-care tests, etc... due to the low specificity of these tests and the possibility of cross-reactions with other parasites that infect dogs, not only trypanosomatids. The use of 2 serological tests is recommended by WHO and PAHO and also applies to the diagnosis of infection in animals for several diseases, including leishmaniasis. Authors should perform a second serological test, and I would recommend a commercial point-of-care test or an ELISA with recombinant L. infantum-specific proteins before defining infections. The article must be rejected and resubmitted after further experiments and analysis.
Moreover, in a future new submission, authors should be careful in the discussion because they compare studies that were performed using different diagnostic methodologies without proper discussion. Studies with different methodologies are not comparable or are only comparable when added to a thorough discussion.
Minor editing of English language required